# Exploring the Neuroprotective Potential of N-Methylpyridinium against LPS-Induced Neuroinflammation: Insights from Molecular Mechanisms

**DOI:** 10.3390/ijms25116000

**Published:** 2024-05-30

**Authors:** Laura Giannotti, Benedetta Di Chiara Stanca, Francesco Spedicato, Eleonora Stanca, Fabrizio Damiano, Stefano Quarta, Marika Massaro, Luisa Siculella

**Affiliations:** 1Department of Experimental Medicine, University of Salento, 73100 Lecce, Italy; eleonora.stanca@unisalento.it; 2Department of Biological and Environmental Sciences and Technologies (DiSTeBA), University of Salento, 73100 Lecce, Italy; benedetta.dichiara@unisalento.it (B.D.C.S.); francesco.spedicato@unisalento.it (F.S.); fabrizio.damiano@unisalento.it (F.D.); 3Institute of Clinical Physiology (IFC), National Research Council (CNR), 73100 Lecce, Italy; stefanoquarta@cnr.it (S.Q.); marika.massaro@cnr.it (M.M.)

**Keywords:** U87MG, neuroinflammation, LPS, NF-κB, NMP

## Abstract

N-methylpyridinium (NMP) is produced through the pyrolysis of trigonelline during the coffee bean roasting process. Preliminary studies suggest that NMP may have health benefits, thanks to its antioxidant properties. Based on this background, the aim of this study was to evaluate whether NMP could have a protective effect against LPS-induced neuroinflammation in human glioblastoma cells (U87MG). With this aim, U87MG cells were pre-treated with NMP (0.5 μM) for 1 h and then exposed to LPS (1 μg/mL) for 24 h. Our findings show that NMP attenuates LPS-induced neuroinflammation by reducing the expression of pro-inflammatory cytokines, such as IL-1β, TNF-α and IL-6, through the inhibition of the NF-κB signaling pathway, which is critical in regulating inflammatory responses. NMP is able to suppress the activation of the NF-κB signaling pathway, suggesting its potential in preventing neuroinflammatory conditions. These outcomes support the notion that regular consumption of NMP, possibly through coffee consumption, may offer protection against neuroinflammatory states implicated in neurological disorders.

## 1. Introduction

The major manifestations of acute inflammatory pathologies involve cellular and molecular processes that are generally transient. Failure to resolve these processes can lead to prolonged inflammation, with continuous recruitment of inflammatory cells to the damaged site, due to failure of macrophages to switch to an anti-inflammatory phenotype, and lack of partial tissue regeneration [1].

Neuroinflammation is an inflammatory process that occurs in nervous tissue and involves cells of the central nervous system (CNS) and immune system [2]. Glial cells (microglia and astrocytes) are considered the innate immune cells of the CNS, synthesizing and secreting various mediators to maintain brain tissue homeostasis during physiological and pathophysiological responses [3].

Consequently, neurotoxic insults, including drugs and pathogens, can induce a reactive gliosis process that promotes the release of pro-inflammatory cytokines, such as tumor necrosis factor-alpha (TNF-α) and interleukin-1 beta (IL-1β), leading to morphological, metabolic and molecular changes [4,5,6]. Therefore, chronic neuroinflammation is associated with the development of several neurodegenerative diseases and neurological disorders [2].

The lipopolysaccharide (LPS)-induced inflammation model is widely used to study neuroinflammation because neuronal cells express toll-like receptor 4 (TLR 4), the major receptor for LPS. Its activation leads to TLR4 receptor dimerization and recruitment of intracellular adaptor proteins, activating the nuclear factor kappa B (NF-κB p65) pathway and inducing the release of major pro-inflammatory cytokines, such as IL-1β, TNF-α and interleukin-6 (IL-6) [7,8]. In addition, in vivo studies have shown that LPS administration is responsible for an astroglial response with upregulation of the acidic gliofibrillary protein (GFAP) [9].

Lately, significant emphasis has been placed on the exploration of naturally existing substrates, due to their diverse array of biological effects. Coffee is rich in bioactive components such as caffeic acid, chlorogenic acids and trigonelline; these components depend on the species of coffee, the roasting of the beans or the method of preparation [10].

N-methylpyridinium (NMP) is a coffee phytochemical produced during coffee bean roasting. This process leads to the degradation of the trigonelline molecule and the release of by-products such as NMP, which is known for its cytoprotective and antioxidant effects [11,12].

Lang et al. demonstrated that moderate consumption of coffee is associated with a reduced risk of developing pathological dysfunctions, involving oxidative stress and neuroinflammatory processes, such as Parkinson’s and Alzheimer’s disease [13,14].

In vivo studies have shown that NMP appears early in plasma but has a rather short half-life. Nevertheless, it can activate cellular detoxification mechanisms, including those based on the Nrf2/ARE pathway and on the expression and the activity of phase II enzymes [11]. Therefore, the aim of this study was to investigate the potential beneficial effects of NMP in inhibiting LPS-induced neuroinflammation in human glioblastoma cells (U87MG) and to analyze the molecular mechanisms involved.

## 2. Results

### 2.1. Effects of NMP on Cell Viability

Cell viability and growth curve analyses were performed through MTT assay and cell count, after treating U87MG cells with NMP at different concentrations (0.01–0.1–0.5–1–5–10 μM) for 24 h. No cytotoxicity was observed at concentrations in the range of 0.01–1 μM but, as the concentration of NMP increased (5, 10 μM), the treatment determined a reduction in cell viability and cell growth (*p* < 0.001, Figure 1A,B). Therefore, the NMP concentration of 0.5 μM was considered optimal for the following experiments.

Inflammation in U87MG cells was induced by using 1 μg/mL LPS for 24 h, as reported in previous studies [8]. The viability of U87MG cells pre-treated with NMP (0.5 μM for 1 h) and then stimulated with 1 μg/mL LPS for 24 h, this condition indicated by LPS-NMP, has been evaluated. The results (Figure 1C) indicated that LPS treatment caused a decrease in cell viability compared to the control cells (CTR), while the LPS-NMP condition showed an increase in U87MG cell viability. No change in cell viability was observed in cells treated with NMP alone compared to CTR.

### 2.2. NMP Inhibits LPS-Induced Inflammation in Reactive Astrocytes

Several data show that astrocytes, after administration of inducers such as LPS, show increased expression of the GFAP, which is considered a marker for reactive astrocytes [9]. Therefore, U87MG cells were treated with NMP (0.5 μM) for 1 h, followed by LPS stimulation (1 μg/mL) for 24 h before evaluating GFAP expression. As shown in Figure 2A, LPS treatment significantly increased GFAP protein expression compared to the control, whereas protein levels were suppressed by the combined treatment with NMP and LPS (LPS-NMP) compared to the LPS condition. Furthermore, image analysis by immunofluorescence staining was performed to examine the changes of GFAP expression in U87MG cells in all the conditions analyzed. Cells were stained with DAPI to highlight the nuclei and labeled with anti-GFAP antibody. Compared to the control, LPS-stimulated cells showed increased immunoreactivity suggesting the induction of cell inflammation, whereas pre-treatment with 0.5 μM NMP in LPS-treated cells showed a reduction in GFAP immunoreactivity in U87MG cells (Figure 2B). These data were statistically confirmed by relative fluorescence intensity analysis (Figure 2C).

### 2.3. NMP Regulates the Expression of Pro-Inflammatory Cytokines in U87MG Cells

Pro-inflammatory cytokine mRNA levels were assessed by RT-PCR to determine whether NMP could regulate LPS-induced pro-inflammatory activation. As shown in Figure 3A–C, LPS significantly increased the mRNA levels of IL-1β, TNF-α and IL-6 compared to the control, whereas combined treatment with LPS and NMP inhibited the expression of LPS-induced IL-1β, TNF-α and IL-6. These results demonstrated that NMP significantly inhibited neuroinflammatory responses by reducing the expression of pro-inflammatory cytokines in U87MG cells.

### 2.4. NMP Acts at the Level of the NF-κB Signaling Pathway in U87MG Cells

It has been reported that LPS induces activation of the NF-κB pathway resulting in the production of pro-inflammatory cytokines [8]. Therefore, the neuroprotective effects of NMP on the activated NF-κB pathway after LPS treatment were evaluated.

Western blotting results showed that the administration of LPS alone increased the phosphorylation of IκBα (p-IκBα) and NF-κB p65 (p-NF-κB) in U87MG cells compared to the control; instead, the pre-treatment with NMP significantly reduced the expression of p-IκBα and p-NF-κB in cells stimulated with LPS (Figure 4A,B).

A further investigation carried out in this study was to examine the cytoplasmic–nuclear translocation of the p65 subunit of NF-κB in U87MG cells treated with and without NMP after stimulation with LPS. Therefore, Western blotting was performed by analyzing NF-κB p65 content in the cytosolic and nuclear protein extracts from untreated- and treated-U87MG cells. The results indicated that LPS induced the NF-κB signaling pathway activation as suggested by the significant nuclear translocation of NF-κB p65 (Figure 5A). Conversely, the pre-treatment with 0.5 μM NMP effectively prevented this process (Figure 5B), suggesting a clear inhibitory effect on the LPS-induced NF-κB pathway (*p* < 0.01). These results were confirmed by immunofluorescence, which showed a decrease and an increase in NF-κB in the cytoplasm and the nuclei, respectively, in LPS-induced cells (Figure 5C).

### 2.5. NMP Inhibits the Transactivation Activity of NF-κB Transcription Factor in LPS-Treated Cells

To investigate whether the treatment with NMP and LPS could affect the transactivation activity of NF-κB, two copies of the NF-κB binding motifs were inserted upstream of the firefly luciferase (FL) cistron of the pGL3 prom plasmid to create the pGL3-2xNF-κB (Figure 6A). This construct was used to transiently transfect U87MG cells, which, after transfection, were treated with NMP and/or LPS for another 24 h. In this experimental condition, the expression of firefly luciferase (FL) depends on the activation of NF-κB triggered by LPS stimulation.

As shown in Figure 6B, all the treatments did not produce any changes in the FL activity of cells transfected with the empty vector pGL3prom. On the contrary, transfection with pGL3-2xNF-κB resulted in significant variations in the FL activity: in fact, while the only NMP treatment did not show any difference with respect to untreated cells, the treatment with LPS caused a 2.4-fold increment in FL activity with respect to CTR. Interestingly, the co-treatment with NMP and LPS (LPS-NMP) determined a 2.75-fold decrease in FL activity with respect to LPS-treated cells.

## 3. Discussion

It has been reported that regular consumption of coffee has protective effects against the development of neurodegenerative conditions [14]. Despite past negative associations of coffee consumption, more recent studies have highlighted its potential benefits [15,16]. During the coffee bean roasting process, secondary products such as nicotinic acid and NMP compounds are formed [17]. In vivo studies have shown that NMP exhibits rapid plasma uptake, indicating rapid intestinal absorption. Its daily plasma concentration is between 0.1 and 0.4 μmol L^−1^ [18]. Recently, the molecule NMP has shown health benefits, including antioxidant properties [11], vaso-protection [19], insulin sensitization [20] and anti-inflammatory effects [15].

The release of pro-inflammatory cytotoxic factors may contribute to the uncontrolled activation of glia, which is the basis of several neurodegenerative diseases [21,22].

Recently, Murata et al. showed, using BV-2 microglial cells, that both decaffeinated and regular coffee caused a reduction in the inflammatory response induced by LPS, mediated by inhibition of NF-κB. They suggested that coffee consumption suppresses neuroinflammation and hypothesized that it may prevent the onset of various neurodegenerative diseases [23].

Therefore, based on these published data, the aim of this study was to analyze the anti-inflammatory and neuroprotective effects of NMP, hypothesizing that this molecule may have the ability to counteract LPS-induced neuroinflammation in human glioblastoma cells (U87MG). In the present study, the exposure of U87MG cells to LPS treatment alone (1 μg/mL) for 24 h significantly reduced cell proliferation and increased GFAP levels; conversely, pre-treatment with NMP (0.5 μM), followed by LPS treatment, showed improved cell proliferation and increased GFAP expression levels. When glial cells are excessively activated, there is a continuous release of pro-inflammatory cytokines and other potentially cytotoxic molecules such as IL-1β, TNF-α and IL-6, which contribute to progressive neuronal damage [8].

This study showed that the mRNA levels of pro-inflammatory cytokines in U87MG cells were increased by LPS stimulation. However, the expression of IL-1β, TNF-α and IL-6 was significantly reduced in U87MG cells pre-treated with NMP and then stimulated with LPS, suggesting that the anti-inflammatory effects of NMP were likely related to the inhibition of pro-inflammatory cytokines release. Furthermore, this study showed that pre-treatment with NMP significantly inhibited the activation of the NF-κB pathway after LPS stimulation of U87MG cells. The NF-κB pathway plays a critical role in regulating inflammatory responses to extracellular stimuli and is therefore considered an important target for anti-inflammatory molecules [24]. The NF-κB dimer (p65/p50) is preferentially localized in the cytoplasm, where it binds to the inhibitory protein IκB. Upon stimulation by pro-inflammatory molecules such as LPS, the IκB protein is rapidly phosphorylated and degraded by the proteasome pathway [25,26,27]. NF-κB, released from the inhibitor, translocates to the nucleus, where it promotes the transcription of target genes for pro-inflammatory cytokines [28].

To further investigate the potential molecular mechanism by which NMP attenuates the production of pro-inflammatory cytokines, activation of the NF-κB pathway was examined. After stimulating U87MG cells with LPS, the activated pathway involved the degradation of IκBα and of its phosphorylated form, as well as the nuclear translocation of NF-κB, which was significantly increased, whereas NMP appeared to suppress LPS-induced NF-κB activation. Therefore, the present results strongly suggest that NMP prevents LPS-induced neuroinflammation and this finding may be related to the inhibition of the NF-κB signaling pathway activation [29]. Previous studies have shown that inhibition of the TLR4/TRAF6 signaling cascades suppresses microglia response to neuroinflammation [30]. In addition, activation of the TLR4 signaling pathway induces NF-κB activation and triggers the release of pro-inflammatory cytokines in astrocytes [31]. Therefore, administration of NMP could have effects on the TLR4/MyD88/TRAF6 signaling pathway, that is upstream on the NF-κB signaling pathway activation.

These results confirm the anti-inflammatory effects of NMP and suggest that consistent consumption could prevent or reduce a neuroinflammatory state that may underlie neurological disorders.

## 4. Materials and Methods

### 4.1. Cell Culture and Treatment

U87MG cells (HTB-14, ATCC, Rockville, MD, USA) were cultured in high glucose DMEM containing 10% (*v*/*v*) fetal bovine serum (FBS), 1% (*v*/*v*) penicillin–streptomycin solution and 2 mM L-glutamine. Cells were incubated in a humidified incubator containing 95% air and 5% CO_2_ at 37 °C. When confluence reached approximately 80%, the cells were maintained under serum-free conditions for pharmacological treatment. Lyophilized powder of LPS (L4516 Sigma-Aldrich, Darmstadt, Germany) was dissolved in 0.9% saline to a stock concentration of 1 mg/mL and then used at a final concentration of 1 μg/mL. NMP (69697 Merk, Darmstadt, Germany), also in powder form, was dissolved in dimethyl sulfoxide (DMSO) to a stock concentration of 5 mM and used at a final concentration of 0.5 μM. The following experimental conditions were used: control (CTR, cells treated with DMSO only), NMP 0.5 μM (NMP), LPS 1 μg/mL (LPS), and co-incubation of NMP (1 h pre-treatment) and LPS (24 h treatment) to assess any neuroprotective effects of NMP against LPS-induced cytotoxicity (LPS-NMP). The experiments performed included a control condition without DMSO and there were no differences compared to the control condition with DMSO.

### 4.2. Cell Viability Assay

The MTT (3-(4,5-dimethylthiazol-2-yl)-2,5-diphenyltetrazolium bromide) assay was performed to assess the viability of U87MG cells after treatment with LPS and/or NMP. The concentration of LPS used was 1 μg/mL for 24 h, based on previous literature [8]. Cells were seeded at a density of 6 × 10^3^ cells/well in a 96-well plate and treated with different concentrations of NMP (0.01, 0.1, 0.5, 1, 5 and 10 μM) for 24 h to assess the cytotoxicity of this molecule. The concentrations used are consistent with those used by Quarta et al. [15] and equivalent to those reported in vivo [21]. After determining the optimal concentration of 0.5 μM NMP, U87MG cells were seeded as previously described and treated according to the previously described experimental conditions. Then, 20 μL MTT solution (10 mg/mL) was added to each well and incubated at 37 °C for 4 h. After the reaction, the absorbance was measured using a plate reader at 570 nm.

### 4.3. Cell Counting

Cells were seeded at a density of 5 × 10^4^ cells/well in a 12-well plate and treated with different concentrations of NMP (0.01, 0.5, 1, 5 and 10 μM) for 24 h. At the end of the treatment time, the cells were incubated with 0.25% trypsin for 5 min at 37 °C; then, the trypsin action was blocked with DMEM high glucose medium with 10% FBS. Finally, 20 μL was taken for cell counting by Bürker chambers counting [32].

### 4.4. Immunofluorescence Assay

U87MG cells were seeded on sterile slides in a 24-well plate at a density of 1 × 10^4^ cells/well. After 24 h, the growth medium was replaced with serum-free medium and the cells were treated according to the experimental conditions described in Section 4.1. After the 24 h treatment, cells were fixed with 4% (*v*/*v*) paraformaldehyde in PBS for 10 min. Cell samples were washed with phosphate-buffered saline (PBS), permeabilized with pH 7.4 PBS containing 0.2% (*v*/*v*) Triton X-100 for 10 min at 37 °C. The cells were then washed in PBS before blocking non-specific binding sites with a 5% BSA solution in PBS for 1 h at room temperature. The cells were incubated overnight at 4 °C with mouse monoclonal anti-GFAP antibody (1:100, sc-166481) and in parallel with rabbit monoclonal anti-NF-κB p65 antibody (1:100, #8242) prepared in a 0.5% BSA solution in PBS. The samples were then washed with PBS and incubated with AlexaFluor FITC-conjugated goat anti-mouse secondary antibody (for the first one) or with AlexaFluor R-phycoerythrin-conjugated goat anti-rabbit secondary antibody (for the second one) (A90-138F and A120-116PE, respectively) for 1 h in the dark at room temperature. In addition, a negative control was performed by incubating the U87MG cells without primary antibody. Finally, the samples were mounted and stained with DAPI (0100-20 Southern Biotech, Birmingham, AL, USA) on a microscope slide and the immune complexes were visualized by confocal microscopy (LSM 900 ZEISS, Oberkochen, Germany).

### 4.5. Real-Time PCR

U87MG cells were seeded in 12-well plates at a density of 9 × 10^4^ cells/well and treated according to the experimental conditions described in Section 4.1. RNA was extracted from each condition using Trizol (T9424 Sigma, Merck Life Science S.r.l., Milan, Italy) following the manufacturer’s instructions. Reverse transcription was carried out with 1 μg of total RNA, random primers and MultiScribe^®^ Reverse Transcriptase (CA94404 Applied Biosystems, Monza, Italy) according to the manufacturer’s guidelines, in a 20 μL reaction. Quantitative analysis of gene expression was conducted on a CFX Connect Real-Time system (Bio-Rad, Hercules, CA, USA) utilizing SYBR Green technology (FluoCycle-Euroclone, Milan, Italy). The Gapdh gene was used as an internal control for normalization. The specificity of the PCR products was confirmed by melt curve analysis and the reactions were performed in triplicate on three independent sets of RNA. The primer sequences used are listed in Table 1.

### 4.6. Western Blotting Analysis

U87MG cells were seeded at a concentration of 1.0 × 10^6^ cells per 100 mm dish for each experimental condition (see Section 4.1). To obtain total cellular protein extracts for Western blot analysis, cells were scraped in the following buffer: 20 mM Tris-HCl (pH 8.0), 420 mM NaCl, 2 mM EDTA, 2 mM Na_3_VO_4_ and 1% (*v*/*v*) Nonidet P-40, supplemented with a cocktail of protease inhibitors. Cell lysates were subjected to freeze/thaw cycles and then centrifuged at 10,000 rpm for 10 min at room temperature. The supernatant was then collected and frozen until use.

The nuclear and cytosolic proteins were obtained from the cell pellets with the cytoplasmic and nuclear extraction buffer containing protease and phosphatase inhibitors (78833 Thermo Fisher, Waltham, MA, USA). After centrifugation at 15,000 rpm for 30 min at 4 °C, the supernatant was collected. Protein extract concentrations (total, cytosolic and nuclear) were determined using the Bio-Rad protein assay kit. Lyophilized bovine serum albumin (BSA) was used as a standard. Equal amounts of protein were denatured at 96 °C for 5 min and separated on 10% (*w*/*v*) SDS gels. The separated proteins were then electrophoretically transferred to a nitrocellulose membrane (Pall, East Hills, NY, USA). Equal loading of proteins was confirmed by Ponceau S staining. The membrane was blocked with 2.5% (*w*/*v*) non-fat dry milk in buffered saline for 1 h at room temperature. The blots were then incubated overnight at 4 °C with specific primary antibodies at 1:1000 dilution. The primary antibodies were Anti-GFAP (sc-166481), NF-κB p65 (#8242), p-NF-κB p65 (sc-135648), p-IκBα (sc-8404), IκBα (sc-1643), anti-β-actin (sc-47778) and anti-lamin A/C (#4777).

Immunocomplexes were detected using appropriate secondary antibodies conjugated to peroxidase (anti-mouse A90-116P and anti-rabbit A120-101P) for 1 h at room temperature, and immunoreactive bands were detected using an enhanced chemiluminescence detection kit #1705061). Densitometric analysis was performed on the blots using the ChemiDoc MP imaging system (Bio-Rad, Hercules, CA, USA).

### 4.7. Cell Transfection and Luciferase Assay

The pGL3-2xNFκB reporter plasmid was obtained by cloning a DNA fragment containing 2 binding motifs for NF-κB transcription factor into the pGL-3 promoter (Promega, Madison, WI, USA) at the KpnI and BglII sites. The DNA fragment was obtained by annealing two oligonucleotides: 5′-CGGGAATTTCCGGGAATTTCCCGGGAATTTCCGGGAATTTCCA-3′ and 5′-GATCTGGAAATTCCCGGAAATTCCCGGGAAATTCCCGGAAATTCCCGGTAC-3′.

For transient transfection, 5 × 10^5^ cells were seeded into 12-well plates 24 h before transfection. U87MG cells were transiently transfected with luciferase reporter construct. Transfection was carried out by using Lipofectamine™ 3000 Transfection Reagent (L3000001 Invitrogen, Waltham, MA, USA) following the manufacturer’s recommendations. After 24 h transfection period, in the medium were added DMSO, NMP, LPS, or NMP and LPS, in the same concentrations as described in paragraph 4.1. After the treatments, cells were lysed, and firefly luciferase activity was measured using Luciferase Assay System (E1500 Promega, Madison, WI, USA). Control experiment was performed using the empty pGL3prom vector.

For the transfection normalization, pcDNA3.1/His/lacZ was used, which encodes for a *β*-galactosidase. Variances in the *β*-galactosidase activity between control and treated-cells were statistically insignificant, confirming that the experimental conditions had no effect on the *β*-galactosidase expression. Luciferase activity values were normalized with respect to the protein concentration.

### 4.8. Statistical Analysis

The data underwent analysis using one-way analysis of variance (ANOVA) followed by the Bonferroni/Dunn post hoc test. Experiments were conducted independently at least three times and results were expressed as mean ± SD. Statistical analysis was carried out utilizing GraphPad Prism 8.0.2 software, with statistical significance defined as *p* < 0.05.

## 5. Conclusions

In conclusion, this study provided highlights on the potential health benefits of NMP and, in particular, its ability to attenuate LPS-induced neuroinflammation in human glioblastoma cells (U87MG). This effect is attributed to NMP inhibition of pro-inflammatory cytokine expression and of the NF-κB signaling pathway activation. These results suggest that regular consumption of NMP, possibly through coffee consumption, could help to prevent or to alleviate neuroinflammatory conditions associated with neurological disorders, thanks to its anti-inflammatory and neuroprotective properties.

## Figures and Tables

**Figure 1 ijms-25-06000-f001:**
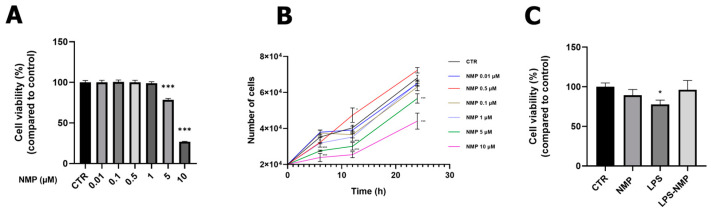
Effects of NMP treatment on LPS-mediated cell viability. U87MG cells were treated with NMP at 0.01–0.1–0.5–1–5–10 μM for 24 h. Viability was determined by MTT assay (**A**) and cell count (**B**). (**C**) U87 cells were pre-treated with NMP (0.5 μM) for 1 h and then incubated with LPS (1 μg/mL) for 24 h. Cell viability values were expressed as percentages relative to control cells. Results are expressed as mean ± standard deviation (SD). Experiments were repeated three times independently (n = 3). * *p* < 0.05 and *** *p* < 0.001 compared with control.

**Figure 2 ijms-25-06000-f002:**
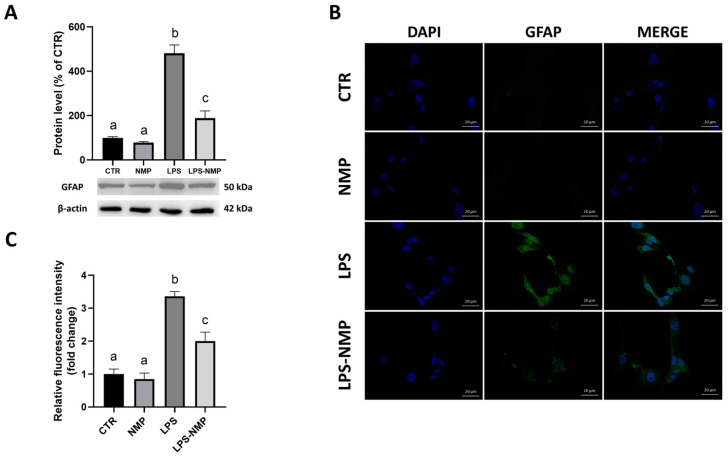
Effects of NMP on LPS-mediated cell activation. (**A**) GFAP protein expression in U87MG cells was determined by densitometric analysis of bands from Western blot. For analysis of GFAP protein expression, values represent relative optical density after normalization to β-actin expression. All values are expressed as mean ± SD (n = 3 replicates) and samples bearing different letters differ significantly (*p* < 0.05). (**B**) Representative immunofluorescence images of GFAP (green) in LPS-stimulated U87MG cells; instead, the nucleus (blue) was stained with DAPI. The scale bar is 20 μm. (**C**) Relative optical density analysis of immunofluorescence images has been determined with Fiji 2.15.0.

**Figure 3 ijms-25-06000-f003:**
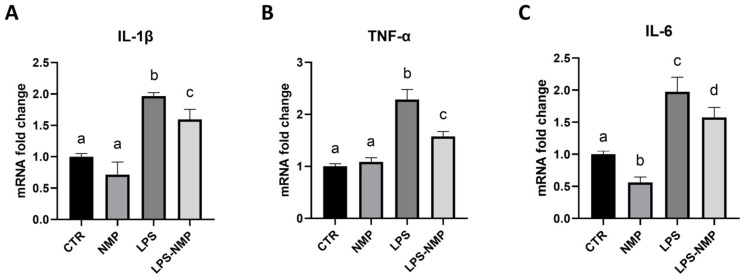
NMP decreased the expression of pro-inflammatory cytokines. The mRNA expression levels of IL-1β (**A**), TNF-α (**B**) and IL-6 (**C**) were decreased after pre-treatment with NMP (0.5 μM) for 1 h, followed by treatment with LPS (1 μg/mL) for 24 h. GAPDH was detected as an internal standard. Results are expressed as mean ± SD; experiments were repeated three times independently (n = 3) and samples bearing different letters differ significantly (*p* < 0.05).

**Figure 4 ijms-25-06000-f004:**
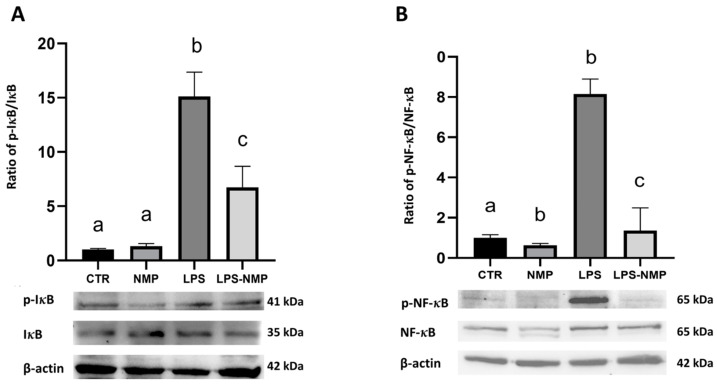
NMP blocked LPS-induced activation of NF-κB p65 in U87MG cells. The ratio of protein expression of: (**A**) p-IκBα/IκBα and (**B**) p-NF-κB p65 and NF-κB p65 analyzed by Western blotting. Results are expressed as mean ± SD; experiments were repeated three times independently (n = 3) and samples bearing different letters differ significantly (*p* < 0.05). β-actin was used as an internal standard.

**Figure 5 ijms-25-06000-f005:**
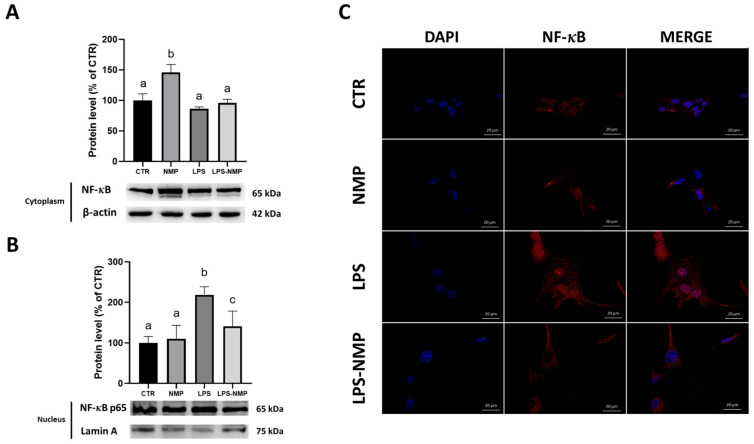
Cytoplasmic-nuclear translocation of NF-κB p65 in U87MG cells. Cytoplasmic (**A**) and nuclear (**B**) NF-κB p65 were determined by Western blotting. β-actin and laminin a/c were used as internal controls for cytoplasmic and nuclear extracts, respectively. Results are expressed as mean ± SD. Experiments were repeated in triplicate independently (n = 3) and samples bearing different letters differ significantly (*p* < 0.05). (**C**) Representative immunofluorescence images of NF-κB p65 (red) in LPS-stimulated U87MG cells. The nucleus (blue) has been highlighted with DAPI. Scale bar is 20 μm.

**Figure 6 ijms-25-06000-f006:**
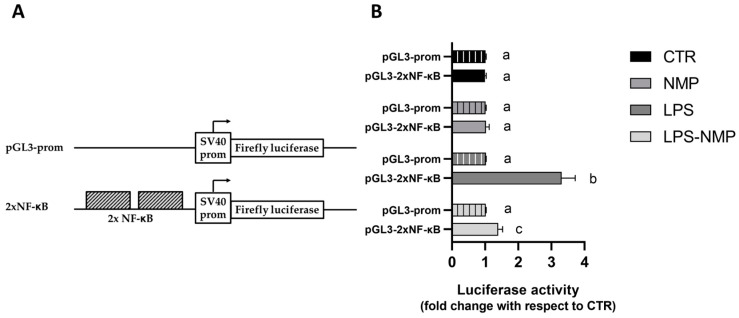
Transactivation activity of NF-κB transcription factor in LPS-treated cells. Luciferase in vitro assay performed with the pGL3-prom vector and pGL3-2xNF-κB construct (**A**). (**B**) The histograms report the relative luciferase activity induced by the pGL3-prom (histograms with vertical lines) and pGL3-2xNF-κB constructs (histogram without vertical lines) transiently transfected into U87MG cells. In both cases, the transfected cells were treated with NMP, LPS, or NMP and LPS (LPS-NMP) for 24 h. The luciferase activity values were reported as fold induction with respect to the activity determined in the control, represented by untreated cells transfected with pGL3-prom or pGL3-2xNF-κB vector. Values were reported as mean ± SD of three independent assays, each performed in triplicate. Samples bearing different letters differ significantly (*p* < 0.05).

**Table 1 ijms-25-06000-t001:** Oligonucleotides used for real-time PCR analysis.

Gene Name	Sequences (5′-3′)
Gapdh	F: ATGGCCTTCCGTGTCCCCAC R: ACGCCTGCTTCACCACCTTC
TNF-α	F: CCCGAGTGACAAGCCTGAG R: GATGGCAGAGAGGAGGTTGAC
IL-1β	F: CTGTCCTGCGTGTTGAAAGA R: AGTTATATCCTGGCCGCCTT
IL-6	F: ACAGCCACTCACCTCTTCAG R: CCATCTTTTTCAGCCATCTTT

## Data Availability

Data is contained within the article.

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
