# Peer review of "Exploring the Neuroprotective Potential of N-Methylpyridinium against LPS-Induced Neuroinflammation: Insights from Molecular Mechanisms"

_ijms, 2024, doi:10.3390/ijms25116000_

Round 1
Reviewer 1 Report
Comments and Suggestions for Authors
Overview of the manuscript
The manuscript focuses on the evaluating the protective effect against neuroinflammation of the
N-methylpyridinium (NMP), a compound produced following the coffee bean roasting process. The authors perform their experimental plan using a cell line of glioblastoma and the experimental model of LPS-induced inflammation. The authors found that NMP attenuates LPS-induced neuroinflammation by reducing the expression of pro-inflammatory cytokines, through the inhibition of the NF-κB signalling pathway. The authors conclude on the protection against neuroinflammatory states implicated in neurological disorders offered by intake of NMP by coffee consumption.
GENERAL COMMENT
The work is interesting, and the experimental plan is well executed and consistent with the aim of the article. The applied methodologies are appropriate and sufficiently investigative for the purpose of the work. However, same results should be better presented and commented. A revision of some citations is suggested.
Specific comments
Abstract
Pag. 1, line 16-17: the sentence is generic and not focused on the topic of your work. Delete or rewrite it.
Pag. 1, line 18: you do not investigate the inflammatory protection on neuronal cells, but the inhibition of inflammatory response on a glioblastoma model. Delete the direct indication to neurons.
Introduction
Pag. 2, line 65: the citation 12, is not appropriate, the citation 14 about the study on dementia should be cited instead. The reference 12 should be cited in the next sentence.
Results
Pag, 6 Fig. 5: the data presented on Fig. 5 are not clear and appear conflicting with the data of Fig. 6. In particular, it can be noted that NMP induce an increase in cytoplasmic NF-kB (Fig.5A). Howis this result compatible with the result of Fig.6B, where the NMP does not elevate the luciferase signal? Explain it and comment.
The immunofluorescence panel (Fig.5C) shows a clear cytoplasmic positive signal in LPS group, higher than that observable in NMP or CTR experimental group, in contrast with western blot data. Furthermore, LPS_NMP experimental group shows an evident positive signal instead of nucleus signal, that seems to conflict with western blot data.
The positive cytoplasmic evidence of NF-kB in the images of CTR and NMP experimental group appears very low, while a better signal should be noted, as indicated by western blot data.
Explain these discordances.
Fig. 6B: make a more differentiated choice of grey scale to highlight the different experimental groups in the histogram.
Discussion
Pag. 7, line 199: the citation of Ref 19 does not seem adequate. The citation seems focused on methodologic and pharmacokinetic study. Choice a more appropriate reference.
Pag.8, line 213: “NMP … improved cell proliferation and GFAP levels”. The sentence is confounding, NMP reduce LPS induced GFAP expression. Furthermore, improved the cell proliferation in a glioblastoma model may not be a positive result. Rewrite the sentence more clearly.
Materials and Methods
Pag. 9, line 267-268: What does optimal effect mean? Here you anticipate your result. The sentence is not clear. Rewrite it.
Pag. 9, line 269-271: It is not clear why you acquired images of individual wells. Explain better the meaning of this procedure.
Author Response
Abstract
1) Pag. 1, line 16-17: the sentence is generic and not focused on the topic of your work. Delete or rewrite it.
Response: According to the reviewer’s comment, we modified the sentence (line 16-17).
2) Pag. 1, line 18: you do not investigate the inflammatory protection on neuronal cells, but the inhibition of inflammatory response on a glioblastoma model. Delete the direct indication to neurons.
Response: We thank the reviewer for the comment and, as suggested, we deleted the indication to neurons (line 18).
Introduction
3) Pag. 2, line 65: the citation 12, is not appropriate, the citation 14 about the study on dementia should be cited instead. The reference 12 should be cited in the next sentence.
Response: According to the reviewer's suggestion, we modified the citations in lines 65 and 68.
Results
4) Pag, 6 Fig. 5: the data presented on Fig. 5 are not clear and appear conflicting with the data of Fig. 6. In particular, it can be noted that NMP induce an increase in cytoplasmic NF-kB (Fig.5A). Howis this result compatible with the result of Fig.6B, where the NMP does not elevate the luciferase signal? Explain it and comment.
Response: We thank the reviewer for the question, and we have explained the mechanism of action of NF-κB, according to which our results seem to be correct. NF-κB is an inducible transcription factor, normally sequestered in the cytoplasm by inhibitory protein (IκB), and thus inactive. When appropriate stimuli occur, such as cytokines or stress inducers, IκB gets phosphorylated and degraded, so NF-κB is able to translocate into the nucleus and exert its anti-inflammatory functions. Based on this mechanism of action, the high cytoplasmic level of NF-κB in the NMP condition (fig 5A) corresponds to low nuclear level (fig 5B), and therefore this does not increase the luciferase activity (fig 6A) because the cytoplasmic form is not active. On the contrary, in the LPS condition, low cytoplasmic and high nuclear levels of NF-κB result in increased luciferase activity.
As we stated in lines 184-186, the expression of firefly luciferase depends on the activation of NF-κB, and on its subsequent nuclear translocation. NF-κB mechanism of action was also reported in the discussion section (lines 245-250).
5) The immunofluorescence panel (Fig.5C) shows a clear cytoplasmic positive signal in LPS group, higher than that observable in NMP or CTR experimental group, in contrast with western blot data. Furthermore, LPS_NMP experimental group shows an evident positive signal instead of nucleus signal, that seems to conflict with western blot data.
Response: We thank the reviewer for this comment. We modified the immunofluorescence panel (Fig.5C), by choosing different image fields of both CTR and NMP conditions, in order to show a signal more in accordance with western blot data. Regarding the LPS-NMP experimental group, the panel shows a more positive signal in cytoplasm than in nucleus, according to western blot data.
6) The positive cytoplasmic evidence of NF-kB in the images of CTR and NMP experimental group appears very low, while a better signal should be noted, as indicated by western blot data.
Response: We thank the reviewer for this comment. We changed the images of CTR and NMP experimental groups.
7) Explain these discordances.
Response: We explained the discordances regarding Fig.5 in the previous two points.
8) Fig. 6B: make a more differentiated choice of grey scale to highlight the different experimental groups in the histogram.
Response: Thanks to the reviewer, we have made the changes to Figure 6B.
Discussion
9) Pag. 7, line 199: the citation of Ref 19 does not seem adequate. The citation seems focused on methodologic and pharmacokinetic study. Choice a more appropriate reference.
Response: In accordance with the reviewer's recommendation, we have replaced citation 19 (line 199) with the already existing reference number 11 (line 405-407) as it is more relevant.
10) Pag.8, line 213: “NMP … improved cell proliferation and GFAP levels”. The sentence is confounding, NMP reduce LPS induced GFAP expression. Furthermore, improved the cell proliferation in a glioblastoma model may not be a positive result. Rewrite the sentence more clearly.
Response: we thank the reviewer and have rephrased the sentence for clarity (lines 217-220).
Materials and Methods
11) Pag. 9, line 267-268: What does optimal effect mean? Here you anticipate your result. The sentence is not clear. Rewrite it.
Response: As suggested, we made the sentence clearer for readability (lines 282-283).
12) Pag. 9, line 269-271: It is not clear why you acquired images of individual wells. Explain better the meaning of this procedure.
Response: Thanks to the reviewer for this note. Images were acquired before performing the MTT test in order to visually demonstrate the non-toxicity of NMP on the U87-MG cell line at the tested concentrations. However, since these images were not included in the present work, we decided to delete the reported sentence (line 292-293).
Reviewer 2 Report
Comments and Suggestions for Authors
The manuscript entitled "Exploring the neuroprotective potential of N-methylpyri-dinium against LPS-induced neuroinflammation: insights from molecular mechanisms" the authors present in an interesting way the role of N-methylpyridine (NMP), a compound produced in the process of roasting coffee beans, as a substance with properties that prevent neuroinflammation caused by the use of lipopolysaccharide (LPS) in the cells of the nervous system.
In the introduction, the Authors briefly explain the pathological significance of inflammation in the nervous system, which may contribute to morphological, metabolic and molecular changes. Then, the authors explain why they were interested in this research topic and set the goal of the research, which is to investigate the potential beneficial effect of NMP on inhibiting LPS-induced neuroinflammation in human glioma cells (U87MG) and analysis of the related molecular mechanisms.
In the experimental part, the Authors of the manuscript established and maintained a cell culture of the U87 MG line of glioblastoma multiforme, and assessed the viability using the MTT test. Immunofluorescence assays, real-time PCR, Western blot analysis, cell transfection, and luciferase assay were also performed.
This study provided the most important information about the potential health benefits of NMP, in particular the possibility of reducing the inflammation caused by LPS. The NMP compound has properties that reduce the expression of cytokines and the activation of the NF-κB signaling pathway.
The results presented by the authors confirm the anti-inflammatory effect of NMP and suggest that consistent consumption may prevent or reduce neuroinflammation, which may underlie neurological disorders such as Parkinson's and Alzheimer's disease. The Authors present the most important information and correlations in a clear and aesthetic way in the attached tables and figures.
In the presented manuscript, the Authors clearly present the methodology and results of the conducted research panel.
The article is focused, timely interesting and substantially detailed. I recommend publication of the work because, in my opinion, the manuscript is interesting for the Readers.
Author Response
Response: We thank the reviewer for analyzing and appreciating our work.
Reviewer 3 Report
Comments and Suggestions for Authors
The goal of this in vitro work was to use the human glioblastoma cell line (U87MG) to investigate the protective effects of N-methylpyridinium (NMP) against LPS-elicited inflammation. The findings are interesting, revealing that NMP inhibited the production of pro-inflammatory cytokines and the NF-κB signaling pathway. Overall, the authors showed that the compound has potential health benefits which could be further investigated due to its anti-inflammatory capacity.
There are some points that I would like to suggest that have room for improvement which are presented below.
- The abstract should contain the concentrations used of N-methylpyridinium (NMP) and lipopolysaccharide (LPS).
- Usually, the keywords should contain terms different from the title to increase the searchability of the work in the future.
- I believe that this sentence “For instance, among the most common flavonoids, quercetin has been demonstrated to significantly attenuate LPS-induced inflammation, thanks to its antioxidative, anti-inflammatory and autophagy-promoting effects [10].” (L. 55-58) does not contribute to the Introduction. Here perhaps the authors should present some study/review on the chemical constituents of coffee, such as caffeine and chlorogenic acid, and their neuroprotective/anti-neuroinflammatory actions. This would better complement what is said in L. 63-65 about coffee consumption and give support to what has been investigated here.
- Figure 1: In the Y axis of figures 1A and 1C change “comparated” to “compared”. Other similar issues should also be checked and corrected.
- Figures 2 and 5: Improve the quality of immunofluorescence images.
- Why there are no error bars for the control groups in the results regarding the expression of pro-inflammatory cytokines (Figure 3) and control groups for western blot analysis (Figure 4)?
- In the Methods section, L. 252-253, the authors stated that “the cells were maintained under serum-free conditions for pharmacological treatment”. I wonder if this does not induce additional stress to cells or was this strategy used because of some serum-binding affinity of the compound?
- The control group (CTR) consisted of cells treated with DMSO only, but what about cells without DMSO (only culture medium) for comparison? Is there evidence to support that the 0.5 μM final DMSO concentration is not cytotoxic to cells?
- Authors need to inform the basis/reference for the concentrations of NMP (0.01, 0.1, 0.5, 1, 5, and 10 μM) and exposure period (24 h) used in this work.
- Finally, why was the human glioblastoma cell line (U87MG) used in this investigation? I would like to suggest the use of another cell line, such as BV-2 microglial cells, that are widely used to assess the neuroinflammatory effects of compounds and neuroimmune responses.
Author Response
The goal of this in vitro work was to use the human glioblastoma cell line (U87MG) to investigate the protective effects of N-methylpyridinium (NMP) against LPS-elicited inflammation. The findings are interesting, revealing that NMP inhibited the production of pro-inflammatory cytokines and the NF-κB signaling pathway. Overall, the authors showed that the compound has potential health benefits which could be further investigated due to its anti-inflammatory capacity.
There are some points that I would like to suggest that have room for improvement which are presented below.
1) The abstract should contain the concentrations used of N-methylpyridinium (NMP) and lipopolysaccharide (LPS).
Response: According to the reviewer’s comment, we modified the sentence (line 19-20).
2) Usually, the keywords should contain terms different from the title to increase the searchability of the work in the future.
Response: We thank the reviewer, and we have made the changes to line 27.
3) I believe that this sentence “For instance, among the most common flavonoids, quercetin has been demonstrated to significantly attenuate LPS-induced inflammation, thanks to its antioxidative, anti-inflammatory and autophagy-promoting effects [10].” (L. 55-58) does not contribute to the Introduction. Here perhaps the authors should present some study/review on the chemical constituents of coffee, such as caffeine and chlorogenic acid, and their neuroprotective/anti-neuroinflammatory actions. This would better complement what is said in L. 63-65 about coffee consumption and give support to what has been investigated here.
Response: According to the reviewer’s comment we amended the text and inserted a new citation (line 56-58, citation 10).
4) Figure 1: In the Y axis of figures 1A and 1C change “comparated” to “compared”. Other similar issues should also be checked and corrected.
Response: We thank the reviewer for this note. We made the change requested (Figure 1) and checked the text.
5) Figures 2 and 5: Improve the quality of immunofluorescence images.
Response: In agreement with the reviewer about the quality of figures 2 and 5, we uploaded the TIFF formats for better quality.
6) Why there are no error bars for the control groups in the results regarding the expression of pro-inflammatory cytokines (Figure 3) and control groups for western blot analysis (Figure 4)?
Response: Thanks to the reviewer for the comment. Initially, the data in these two cases were presented by normalizing the values against the control. Consequently, we didn't report any bar representing the negative control. We corrected the data’s presentation to display error bars in the control group as well.
7) In the Methods section, L. 252-253, the authors stated that “the cells were maintained under serum-free conditions for pharmacological treatment”. I wonder if this does not induce additional stress to cells or was this strategy used because of some serum-binding affinity of the compound?
Response: Thanks for the question. The cells were serum starved to minimize their metabolism without stressing them for such a short time. This technique was used also by Jin H. et al (citation 8).
8) The control group (CTR) consisted of cells treated with DMSO only, but what about cells without DMSO (only culture medium) for comparison? Is there evidence to support that the 0.5 μM final DMSO concentration is not cytotoxic to cells?
Response: Thanks for the observation. We would clarify that the experiments performed included a control condition without DMSO and there were no differences compared to the control condition with DMSO. We added this clarification also in the text (lines 291-292). The decision to use DMSO treatment seemed more appropriate to demonstrate that it was not the vehicle (DMSO) that caused the obtained effect.
9)Authors need to inform the basis/reference for the concentrations of NMP (0.01, 0.1, 0.5, 1, 5, and 10 μM) and exposure period (24 h) used in this work.
Response: We thank the reviewer for this question. In literature, until now, no author has ever tested the effect of the compound NMP on glioblastoma U87-MG cells. For this reason, in our study, an MTT test was first performed to assess cell viability at different concentrations of this compound. The concentrations used are consistent with those used in other cited articles (citation 15) and equivalent to those reported in vivo (citation 20). As suggested by the reviewer, the text has been modified (lines 292-293) to better explain this choice and also appropriate references have been added (citation 15 and 20).
Regarding the exposure period to NMP (24h), this is due to the exposure time to LPS, as reported in the literature and mentioned in line 289. Once the non-toxicity of NMP on U87-MG cells was evaluated, the treatment used to demonstrate its protective effect against LPS-induced cytotoxicity involved pre-treatment with NMP for 1h, followed by combined treatment with LPS for 24h, as already reported in lines 284-285.
10) Finally, why was the human glioblastoma cell line (U87MG) used in this investigation? I would like to suggest the use of another cell line, such as BV-2 microglial cells, that are widely used to assess the neuroinflammatory effects of compounds and neuroimmune responses.
Response: Thanks to the reviewer for suggesting the BV-2 line; in the next work it could be a model for studying the effects of NMP.
Round 2
Reviewer 1 Report
Comments and Suggestions for Authors
The authors have response to all criticism set out in the previous report.
No more concerns
Reviewer 3 Report
Comments and Suggestions for Authors
The authors have improved their manuscript based on the previous suggestions.